# Changes in Knowledge about Umbilical Cord Blood Banking and Genetic Tests Among Pregnant Women from Polish Urban and Rural Areas between 2010–2012 and 2017

**DOI:** 10.3390/ijerph17165744

**Published:** 2020-08-08

**Authors:** Maria Szubert, Malwina Ilowiecka, Jacek Wilczynski, Monika Szpotanska-Sikorska, Cezary Wojtyla

**Affiliations:** 1Clinic of Surgical and Oncologic Gynecology, 1st Department of Gynecology and Obstetrics, Medical University of Lodz, 94-029 Lodz, Poland; malilo1@o2.pl (M.I.); jacek.wilczynski@umed.lodz.pl (J.W.); 2M. Pirogow Teaching Hospital, Wilenska 37 St., 94-029 Lodz, Poland; 31st Department of Obstetrics and Gynecology, Medical University of Warsaw; 1/3 Starynkiewicza Sq, 02-015 Warsaw, Poland; mszpotanska@wp.pl; 4International Prevention Research Institute–Collaborating Centre, State University of Applied Sciences, Kaszubska 16 St., 62-800, Kalisz, Poland; czwo@op.pl; 5Department of Oncologic Gynecology and Obstetrics, Centre of Postgraduate Medical Education, Czerniakowska 231 St., 00-416 Warsaw, Poland

**Keywords:** genetic testing, umbilical cord blood banking, pregnancy, women health

## Abstract

The aim of this study was to evaluate knowledge of umbilical cord blood (UBC) banking and prenatal genetic diagnosis among pregnant women from rural and urban areas, and how this knowledge changed within a five-year period. A survey by questionnaire was conducted between 2010 and 2012, and in 2017 in public hospitals; the study population comprised 6128 women, with 2797 patients from the years 2010–2012 and 3331 from the year 2017. 41% of the studied population declared that they were living in rural areas. In the 2010–2012 period, fewer women from rural areas knew about UBC banking. In 2017 that same relative difference in knowledge persisted, but the percentage of women who now knew about this procedure rose significantly in both studied groups. Prenatal diagnosis was more familiar for urban inhabitants both in 2010 and 2017 but as with the UBC data, a trend of growing awareness was also seen in pregnant women from rural areas. Knowledge of new techniques such as UBC banking and genetic tests has grown among pregnant women during the time frame of our study, but there is still a need to emphasize the benefits of these two possibilities to pregnant women, especially among rural inhabitants.

## 1. Introduction

Dynamic technological changes and development of new, innovative diagnostic methods are discernable in every medical branch, especially in genetics and transplantology [1,2]. Contemporary society has also gained broad access to a variety of medical information through the internet. This unfiltered stream of knowledge can cause difficulties in understanding the relevance and reliability of the acquired information, especially in the medical field. Two recent milestones relating to pregnant women that have added to our knowledge are prenatal genetic diagnosis methods using genetic screening, and umbilical cord blood (UBC) banking of stem cells [3,4]. UBC collecting can be undertaken only once—just after delivery. Stem cell transplants are now used to treat numerous types of immune- and blood-related disorders and genetic diseases [5]. There are, however, significant gaps in parents’ knowledge and awareness of cord blood banking which have been identified in different studies [6]. Women also have limited knowledge about prenatal screening according to Seven et al. [7]. Demographic factors may also affect women’s knowledge about genetic syndromes and prenatal testing [8].

In this study we assessed the knowledge of pregnant women from rural and urban areas regarding UBC banking and genetic testing for medical and scientific use. Our special concern was to identify any changes in knowledge levels during the 5-year timeframe of our study, from the period 2010–2012 to 2017, within rural and urban populations. 

## 2. Materials and Methods 

Our study was conducted by the Chief Sanitary Inspectorate and the Institute of Rural Health in Lublin in the years 2010–2012 and in 2017 across all Polish public hospitals, and involved a randomized group of women and their children. The study analyzed responses to a questionnaire developed within the Pol-PrAMS program (Polish Pregnancy-related Assessment Monitoring System) using the PRAMS (Pregnancy Risk Assessment Monitoring System) model developed in the United States. The Bioethics Committee of the Institute of Rural Health in Lublin approved the study (permission no. 03/2011). 

The participants in the whole study comprised 12,066 pregnant women, including 8625 subjects from the years 2010–2012 and 3441 from 2017. The questionnaire was divided into two parts. The first part (in paper) was completed by the pregnant women and included their personal information and data on pro-health behaviors and the course of pregnancy. The second part was filled in by healthcare professionals based on the medical records of the patients and their children and was gathered after delivery. The results of other parts of the study have been published elsewhere [9]. 

Calculations for the aspect of the study reported here were made after elimination of entries which did not include information on a place of residence, and after correlation of the population characteristics for the years 2010–2012 with that of the 2017 population in terms of age, place and region of residence. The demographic population structure in 2010–2012 was significantly different to that studied in 2017 in terms of age and region of residence. A compromise method of correction was adopted by randomly withdrawing those cases that were overrepresented. The population from the 2010–2012 study was referenced to the population structure from 2017 in terms of age, place of residence (city/village), and region of residence (16 provinces). This data constituted a trivariate table, covering 3331 women subjects from 2017 divided into 3 × 3 × 16 = 144 subgroups. The number of subjects in each subgroup was expressed as a percentage of the total. An analogous table was prepared for the population studied in 2010–2012. Based on the coefficients from the 2017 table, the expected counts in each subgroup for 2010–2012 were calculated, from which the overcounts and undercounts could be derived. The population size was reduced so that there were no undercounts present in the table, only overcounts. This was achieved by reducing the total size of the interviewed population from the 2010–2012 study to 2797 cases. Current and newly expected counts were provided for each subgroup. The required number of cases was then randomly selected from the SPSS database program, containing the data for each subgroup of the 2010–2012 study. This procedure randomly selects an m-element sample of cases from an *n*-element population. There were 144 such procedures separately performed for the subgroups. The adjusted study population of 2797 cases did not differ significantly from the 2017 study population in terms of the unified variable structure. According to this description we analyzed data from 6128 women, comprising 2797 patients from the years 2010–2012 and 3331 from the year 2017. A detailed description of the groups has already been published elsewhere [10]. Flow chart 1 illustrates the whole process of establishing sample size (flow chart near here).

Flow chart 1: Process of inclusion of participants in the study. Number of births in Poland is given for the studied timeframe (for the exact day in each year when procedures of the Pol-PrAMS study were conducted).

Obtained data were compared between years of the survey in women residing in urban and rural areas. The statistical analysis of the collected data was made based on a chi-squared test and Mann–Whitney test using IBM SPSS software version 25 (IBM Corp., Armonk, NY, USA).

## 3. Results

### 3.1. Sociodemographic Characteristics

The largest proportion of the study group surveyed during both periods (2010–2012 and 2017) comprised women aged between 26 and 30 years, and the smallest proportion consisted of women aged over 35 years. Among the respondents, city residents made up 59% of the examined group and women living in rural areas accounted for 41% of the total group (Table 1).

### 3.2. Provision of Specialist Care to the Pregnant Population in Urban and Rural Areas 

Comparing the provision of specialist care to pregnant women in rural and urban areas in the studied time periods of 2010–2012 and 2017, we found that women living in rural areas had gynecological examinations in the 1st, 2nd and 3rd trimester of pregnancy relatively less often than women from urban areas (p < 0.05). Meanwhile, the data from 2017 showed that the general number of medical visits during the 1st, 2nd and 3rd trimester of pregnancy increased for women living in rural and those resident in urban areas when compared with data from the years 2010–2012 (Table 2).

### 3.3. Women’s Knowledge of Umbilical Cord Blood Banking 

Across the whole study period, levels of knowledge about UBC banking among pregnant women resident in urban areas were higher than among those living in rural areas. In 2017, compared with those surveyed in the earlier period, fewer women said that they had not heard of UBC banking at all (1.2% of urban residents and 6.5% (*p* < 0.05) of rural residents respectively). In the answers of urban residents from 2010–2012, 57.2% of the respondents thought that this procedure was useful (vs. 49.4%, *p* < 0.05 of the women from rural areas), whereas 3.1% considered it to be useless (vs. 1.6%, *p* < 0.05 of the women from rural areas) (Figure 1 and Figure 2). In 2017, despite increased awareness about UBC banking, fewer women, from both populations, considered it useful. 

In 2017, there were 4.4% more urban women and 7.6% more rural women who answered that they had no opinion about this procedure (*p* < 0.05).

### 3.4. Women’s Knowledge about Genetic Testing 

In 2017, in both urban and rural populations, the number of women who had never heard of using genetic material for medical or scientific purposes decreased when compared with the answers given in the years 2010–2012 ( Figure 3, Figure 4, Figure 5 and Figure 6). In 2017, among women living in urban areas, there was an 8.3% decrease in patients unconscious of the possibility of genetic testing, whereas among women living in rural areas there was an 8% decrease (*p* < 0.05). On the other hand, the number of women who regarded these tests as necessary increased in both populations in 2017. Among women from urban areas this increase was 4.3%, whereas among women from rural areas it was 6.2% (*p* < 0.05). Moreover, the number of women who answered “I have no opinion on this issue” increased in that same year by 4.3% and 2% in the urban and rural areas respectively (*p* < 0.05). 

A similar trend to that observed in the survey on umbilical cord blood was recorded in the survey on genetic tests. In both survey periods, 2010–2012 and 2017, more women from urban areas had heard of genetic testing compared with those from rural areas. Nevertheless, the data included in Table 3 show that the percentage share of the patients who had never heard of genetic testing is more than 50% of the answers analyzed.

## 4. Discussion

Our assessment of pregnant women’s knowledge and attitudes to cord blood storing is the first in which place of residence and changes in opinions over a 5-year period have been analyzed. It is also the first study conducted in the Polish population concerning both innovative techniques (UBC and genetic testing) that are important for pregnant women and wider society. Genetic testing during pregnancy with the use of cell-free fetal DNA is recommended in clinical obstetric guidelines worldwide [11,12,13,14]. Cell-free fetal DNA represents extracellular DNA which originates from trophoblastic cells that contain the entire fetal genotype. Assessment of cell-free fetal DNA from mother’s blood is useful in the calculation of the risk of congenital genetic syndromes [15]. Women surveyed in our study were asked to give answers about their knowledge of the use of genetic testing in general, and not only about cell-free DNA. Surprisingly more than half of the patients surveyed had not heard about genetic testing at all in 2010–2012. By 2017 that situation had changed, but the knowledge of rural inhabitants was still poorer than that of urban residents. As genetic testing is nowadays the most important part of perinatal counselling, knowledge about this possibility is of significant importance for public health. Women’s rights could be affected if they are not properly tested and counseled for genetic disorders in early pregnancy. 

The first recipient of umbilical cord blood stem cells was a 5-year old boy with severe Fanconi anemia in 1988 [16]. In Poland, a successful frozen and banked umbilical stem cell transplantation in a boy with acute myeloblastic leukemia was performed eight years later [17]. Initially, UBC transplantation was limited to children, and both related and unrelated cord blood transplants have been performed with high rates of success for a variety of hematologic disorders and metabolic storage diseases [16].

Nowadays, UBC is used for adults too, for instance for myelodysplastic syndrome or secondary acute myeloblastic leukemia [18,19]. The use of stem cells from umbilical cord blood has also recently been highlighted in relation to the COVID-19 pandemic. Several case reports about the effective treatment of severe COVID-19 cases have been published, where mesenchymal stem cells (MSCs) have been shown to halt and reverse the cytokine storm [20,21]. 

Studies exploring pregnant women’s and expectant parents’ knowledge and awareness of cord blood donation and banking have already been conducted in 15 countries according to a systematic review by Peberdy et al. [6]. Our study is the first in Poland and was conducted as a part of a larger study on the lifestyle health behaviors of pregnant women. 

According to our data, between 2010–2012 and 2017 the total number of pregnancy monitoring visits during pregnancy significantly increased in both studied populations. This should suggest that the knowledge of pregnant women about new possibilities such as UBC banking and genetic testing ought to have increased too. Polish guidelines for pregnancy management state that gynecologists should inform patients about genetic testing as well as about the possibility of UBC banking or donation [22,23]. While more patients said they had “heard” about UBC banking by 2017 than in the 2010–2012 survey period, regardless of the place of residence, a smaller percentage of patients were convinced about necessity of this procedure. The explanation for this fact could be found in the study by Hatzistilli et al. of health professionals’ knowledge about umbilical cord blood donation. They stated that only 15.6% of the health care providers they surveyed (doctors, nurses, and midwives) declared they were quite well- or well-informed about the collection methods and the uses of stored blood [24]. One can conclude that the less knowledge the health professional has, the poorer the consultation will be, and the rate of UBC banking will remain low. These conclusions are also suggested by others [25,26]. In most countries there are two possibilities for cord blood banking: by donation in public banks or by user-pays storage for private use [27,28]. Lack of exact information about costs, and misunderstanding about who can benefit from UBC banking can, in our opinion, influence the percentage of patients convinced about the value of this procedure; but this aspect was not analyzed in our study. 

## 5. Conclusions

We conclude therefore that it is very important that obstetricians keep up to date with UBC collection and its storage guidelines and financing through educational programs, so that they can share their knowledge with pregnant women, especially in rural areas. The importance of UBC collection and genetic testing should be highlighted during check-up visits from the beginning of pregnancy. Proper genetic testing may have an impact on the rate of severe congenital malformations at birth. Education of nurses, midwives, and doctors could help improve the percentages of pregnant women who understand the value of these two modern techniques. 

## Figures and Tables

**Figure 1 ijerph-17-05744-f001:**
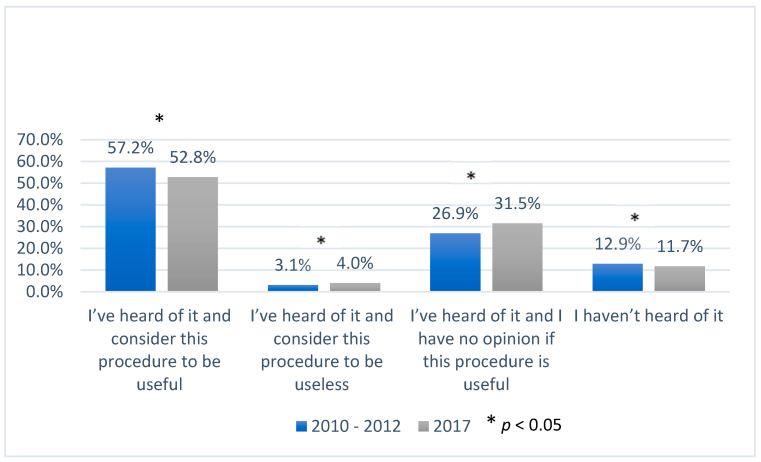
Knowledge and opinions of women from urban areas about umbilical cord blood banking in the years 2010–2012 and 2017; *—*p* < 0.05.

**Figure 2 ijerph-17-05744-f002:**
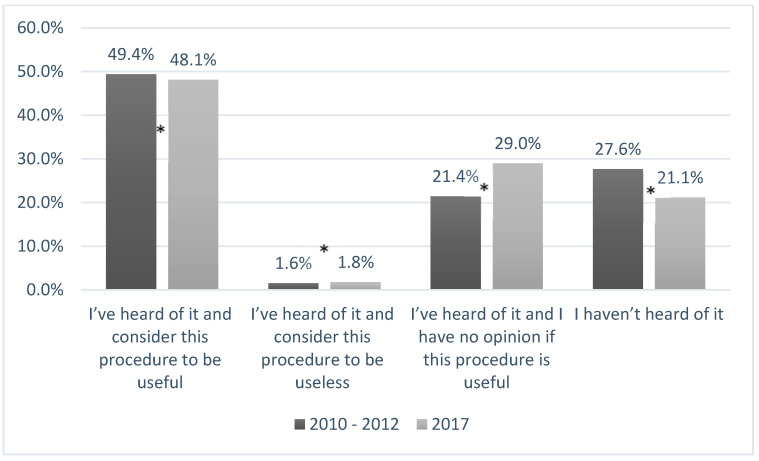
Knowledge and opinions of women from rural areas about umbilical cord blood banking in the years 2010–2012 and 2017; *—*p* < 0.05.

**Figure 3 ijerph-17-05744-f003:**
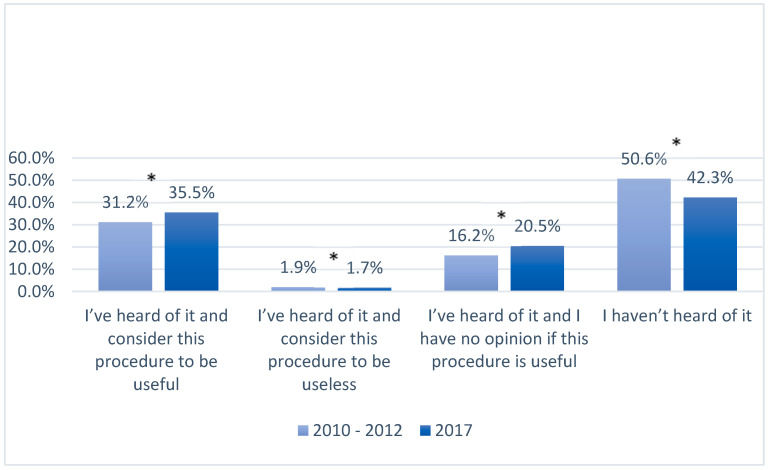
Knowledge and opinions of women from urban populations on using genetic material for medical or scientific purposes in the years 2010–2012 and 2017; *—*p* < 0.05.

**Figure 4 ijerph-17-05744-f004:**
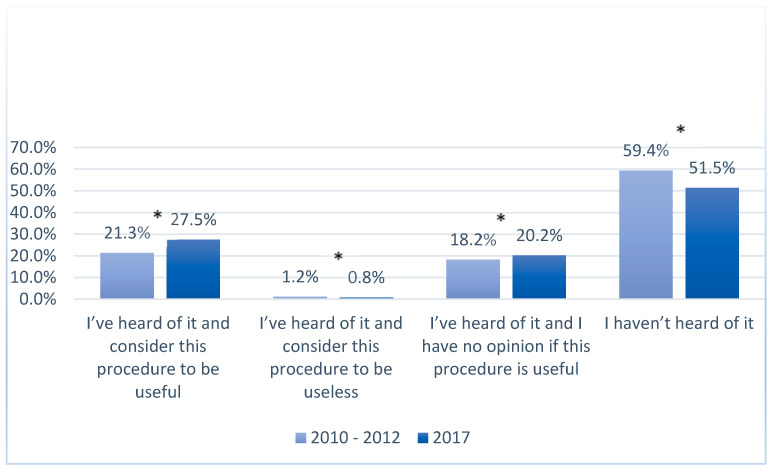
Knowledge and opinions of women from rural populations on using genetic material for medical or scientific purposes in the years 2010–2012 and 2017; *—*p* < 0.05.

**Figure 5 ijerph-17-05744-f005:**
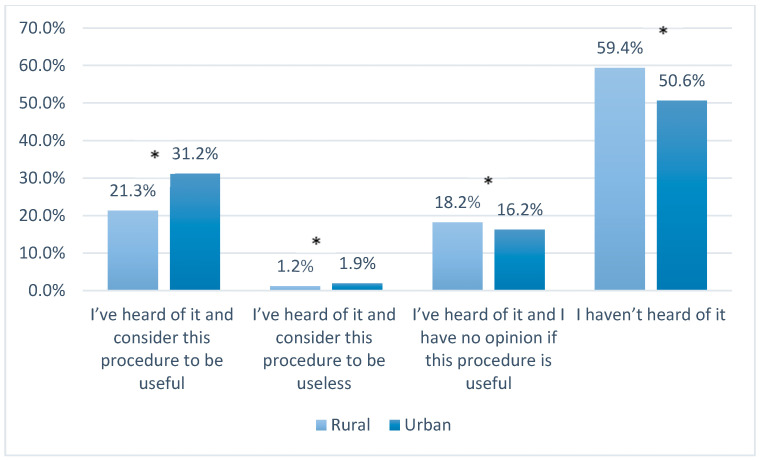
Knowledge and opinions of women from rural and urban populations in the years 2010–2012 on using genetic material for medical or scientific purposes; *—*p* < 0.05.

**Figure 6 ijerph-17-05744-f006:**
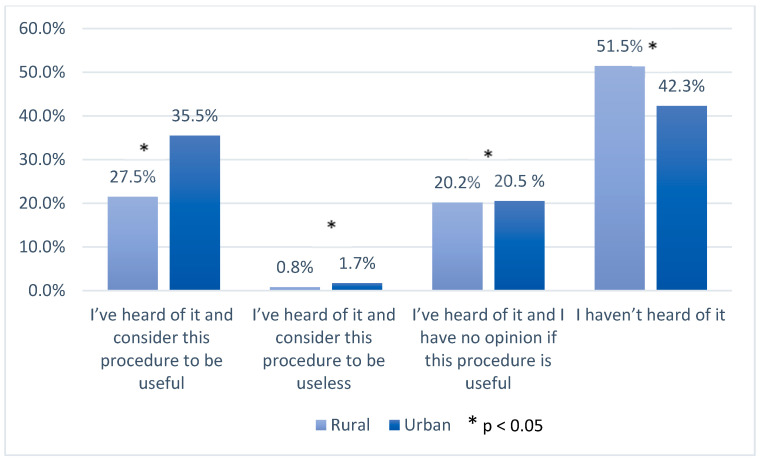
Knowledge and opinions of women from rural and urban populations in 2017 on using genetic material for medical or scientific purposes; *—*p* < 0.05.

**Table 1 ijerph-17-05744-t001:** Characteristics of the study population.

	2010–2012		2017		*p*
	*n*	%	*n*	%	
Age (years)					ns
≤25	532	19.0	627	19.0	
26–30	965	34.6	1139	34.6	
31–35	925	33.1	1073	32.6	
>35	371	13.3	457	13.8	
Place of residence (inhabitants)					ns
City ≥ 100,000	676	24.2	804	24.2	
City < 100,000	967	34.6	1154	34.6	
Rural area	1154	41.2	1373	41.2	
Education					ns
Primary/lower secondary	52	4.8	86	4.5	
Upper secondary/post-secondary	403	37.2	665	34.6	
Higher	596	54.9	1139	59.2	
Other	34	3.1	34	1.7	

ns—non-significant, *p* > 0.05.

**Table 2 ijerph-17-05744-t002:** Frequency of gynecological checkups during pregnancy in urban and rural populations in the years 2010–2012 and 2017.

YEARS 2010–2012							YEAR 2017			
Visit to the Gynecologist			Valid *n*	Mean	SD	Median	Valid *n*	Mean	SD	Median
Visits in the 1st trimester of pregnancy	Place of residence	Rural	722	2.89	1.59	3.28	1187	3.28	1.29	3.00
		Urban	1022	3.20	1.57	3.49	1737	3.49	1.46	3.00
		Total	1744	3.07	1.53	3.40 *	2924	3.40	1.40	3.00 *
Visits in the 2nd trimester of pregnancy	Place of residence	Rural	715	3.43	1.48	3.71	1184	3.71	1.34	3.00
		Urban	1019	3.60	1.88	3.87	1728	3.87	1.49	3.00
		Total	1734	3.53	1.63	3.80 *	2912	3.80	1.43	3.00 *
Visits in the 3rd trimester of pregnancy	Place of residence	Rural	715	4.16	2.09	4.54	1176	4.54	1.96	4.00
		Urban	1018	4.40	2.22	4.79	1708	4.79	2.05	4.00
		Total	1733	4.30	2.12	4.69 *	2884	4.69	2.02	4.00 *

*—*p* < 0.05 (Mann–Whitney test).

**Table 3 ijerph-17-05744-t003:** Knowledge and opinions of all examined women on using genetic material for medical or scientific purposes.

Knowledge of Using Genetic Material from Blood		Year of Study			
		(1) 2010–2012		(2) 2017	
		Number	% of Valid *n* in Column	Number	% of Valid *n* in Column
Have you heard of using genetic material for medical or scientific purposes?	1. I have heard of it and consider this procedure to be useful	500	27.1%	1041	32.2%
	2. I have heard of it and consider this procedure to be useless	30	1.6%	44	1.4%
	3. I have heard of it and I have no opinion whether this procedure is useful	315	17%	659	20.4%
	4. I have not heard of it	1003	54.3%	1488	46%
	In total	1848	100%	3232	100%

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
