# Peer review of "Changes in Knowledge about Umbilical Cord Blood Banking and Genetic Tests among Pregnant Women from Polish Urban and Rural Areas between 2010–2012 and 2017"

_ijerph, 2020, doi:10.3390/ijerph17165744_

Round 1

Reviewer 1 Report

Dear Authors,

In my opinion, the article is well designed and presents interesting results worth to be published in IJERPH. However, I have some minor comments. I wonder why the exclusion rates in 2010-2012 cohort were so high in comparison to 2017? Please consider adding flow chart describing study population. Did two cohorts differ in term of parity? It would also be interesting to test whether the knowledge of genetic tests and umbilical cord blood banking was dependent on maternal age.  

Author Response

D

Reviewer 1:

In my opinion, the article is well designed and presents interesting results worth to be published in IJERPH. However, I have some minor comments. I wonder why the exclusion rates in 2010-2012 cohort were so high in comparison to 2017? Please consider adding flow chart describing study population. Did two cohorts differ in term of parity? It would also be interesting to test whether the knowledge of genetic tests and umbilical cord blood banking was dependent on maternal age.  

Dear Reviewer,

Thank you for your rating of our manuscript. I explained the exclusion rates in lines 69-86, it was also the response for the Reviewer 2 so it is marked both – green and blue.

Data on parity was not analysed in these two populations. Both groups were unified according to age, place, region of residence and the dependence between knowledge and the age was not the aim of this particular study.  Flow chart of study population is addedd to the „M&M” section.

Reviewer 2 Report

Paper goal is to  identify  changes in the knowledge levels during the 5-year timeframe (period 2010-2012-2017) within the rural and urban populations in relation to two topics

  1. umbilical cord blood (UBC) banking
  2. using genetic material 130 for medical or scientific purposes

Paper is describing results of large sample of pregnant women investigated across all Polish public hospitals

The problems raised in the publication are of significant importance for public health in Poland. The publication provide valuable information which might be utilized in process of educating pregnant women and post diploma training of health care professionals.

Minor remarks;

  • The description of the study sample should be provide even if it was presented in other population. Clear information should be provided about the number of approached women (eligible) , those who agreed to participate and those included in the final sample (after a process of adjustment.
  • The statistical significance of the examined association should be presented in each table.
  • The implication of the observed trends for reproductive health are worth more in deep discussion. What type of positive outcomes authors expected in relation to observed trends.

Author Response

Dear Reviewer 2:

Minor remarks;

  • The description of the study sample should be provide even if it was presented in other population. Clear information should be provided about the number of approached women (eligible) , those who agreed to participate and those included in the final sample (after a process of adjustment.

Answer: To make the proces of data gathering clear, we created a flow chart with all the above mentioned information. I hope it fulfills all objectives on study sample. Flow chart is marked green as it was also concern of Reviewer 1. Whole proces of adjustment was described in text (lines – 69-86).

  • The statistical significance of the examined association should be presented in each table.

Answer: Each table was modified and statistical significance was added.

  • The implication of the observed trends for reproductive health are worth more in deep discussion. What type of positive outcomes authors expected in relation to observed trends.

Answer: This point of the discussion was further analyzed in lines: 171-174 and underlined in Conclusions.

Reviewer 3 Report

In general I think that provided study present interesting subject. In many aspects, proper and up-to-date knowledge are crucial and necessary to make right choices. The idea of the study is simple, presentation is adequate and the conclusions adhere to the results. But, I feel that according to possessed data there are unrevealed associations which may or even should be shown in revised manuscript. I don't understand why tests like Kruskall-Wallis or Mann-Whitney were not introduced as they may answer the questions which relations/differences are statistically significant. It seems to be really easy and rapid to improve. At present we have rather descriptive statistics without strong statistic tests. 

Additionally, minor flaws in figures. 

Table 1. Misplaced "967" value. 

Table 2. Should be fixed. What are exactly SD values in 2010-2012 ...? In description you use term "comparison" whereas it's not. 

Figures - It's useless to use description above and below the graph, I would prefer to omit this above sentence. 

Table 3. Usage dots instead of comas. 

Missing references, e.g. ACOG 771; 640 - direct. 

Asterisk in affiliations should be attached to first not last author. 

Author Response

Reviewer 3.

In general I think that provided study present interesting subject. In many aspects, proper and up-to-date knowledge are crucial and necessary to make right choices. The idea of the study is simple, presentation is adequate and the conclusions adhere to the results. But, I feel that according to possessed data there are unrevealed associations which may or even should be shown in revised manuscript. I don't understand why tests like Kruskall-Wallis or Mann-Whitney were not introduced as they may answer the questions which relations/differences are statistically significant. It seems to be really easy and rapid to improve. At present we have rather descriptive statistics without strong statistic tests. 

Dear Reviewer, thank you very much for your comments, below you find our answers:

Answer: Mann-Whitney test was used in statistics regarding control visits during pregnancy as it describes non-parametric data. Although Chi-square test (used in other comparisons) would also be proper for this large group. Ommited test was added to the „M&M section and to subheading for table 2.

Additionally, minor flaws in figures. 

Table 1. Misplaced "967" value. - corrected

Table 2. Should be fixed. What are exactly SD values in 2010-2012 ...? In description you use term "comparison" whereas it's not. 

Answer: Thank you for pointing this out. I corrected SD values as mistake occured during coping the table. I replaced term „comparison”.

Figures - It's useless to use description above and below the graph, I would prefer to omit this above sentence. – As suggested – in all figures above description was deleted.

Table 3. Usage dots instead of comas. – corrected; additionaly column „Total” was deleted as it doubled the previous two columns. 

Missing references, e.g. ACOG 771; 640 - direct. – corrected

Asterisk in affiliations should be attached to first not last author. – corrected

Reviewer 4 Report

1) The text indifferently and sometimes ambiguously mentions "UCB storage" and "UCB banking" without specifying if it is a question of donation or private banking: this in the common opinion can be very different, in particular in the answer concerning the usefulness or uselessness of this opportunity. We can refer to the "UCB matter" only in very general terms

In the discussion and conclusions sections this limitation should be better specified

2) In the discussion, in line 153 it is incorrectly indicated "the first recipient of stem cells was a 5-year-old boy ... in 1988": it is necessary to specify "the first recipient of umbilical cord blood stem cells was ..."

3) likewise, in line 154 it is incorrectly referred to as "a successful stem cell transplantation in a boy .." must be specified as "a successful frozen and banked umbilical stem cell transplantation in a boy ..."

4) in figure 6 there is no correspondence between the percentages indicated in the previous figures (3 and 4). On the contrary, in figure 5, the percentages present in figures 3 and 4 are correctly reported. we ask to double check the correctness of figure 6 (genetic material 2017): I expect: "useful" NO 48.1% -52.8%, YES 27.5% -35.5%;
"useless" NO 1.8% -4.0%, YES 0.8% -1.7%, "no opinion" NO 29.0% -31.5%, YES 20.2% -20.5% ; "I haven't heard" NO 21.1% -11.7%, YES 51.4% -42.3%. Check whether the data relating to the "umbilical cord bank" are crossed with the "genetic material" data.

5) It is recommended to review the conclusions in the light of the re-evaluation of the data

Author Response

Reviewer 4:

  • The text indifferently and sometimes ambiguously mentions "UCB storage" and "UCB banking" without specifying if it is a question of donation or private banking: this in the common opinion can be very different, in particular in the answer concerning the usefulness or uselessness of this opportunity. We can refer to the "UCB matter" only in very general terms. In the discussion and conclusions sections this limitation should be better specified

Answer: According to ACOG opinion UCB storage procedure or banking or collecting are eaqual  terms for acquire of umbilical stem cells. There are two ways of storage and handling with USB: public and privat. [https://www.acog.org/clinical/clinical-guidance/committee-opinion/articles/2019/03/umbilical-cord-blood-banking] In Poland there are also two possibilities of UCB storage, none of them was mentioned in the questionnaire in our study. We assessed only the knowledge about existence of this procedure, we did not examine attitude of pregnant women towards one or the other way of storage. Nevertheless I tried to unified meaning of UBS by deleting „storage” . In the discussion section financial circumstances regarding UBS banking has been already described in lines 202-211 but it was added to Conclusion section.

2) In the discussion, in line 153 it is incorrectly indicated "the first recipient of stem cells was a 5-year-old boy ... in 1988": it is necessary to specify "the first recipient of umbilical cord blood stem cells was ..." – I corrected this statement.

3) likewise, in line 154 it is incorrectly referred to as "a successful stem cell transplantation in a boy .." must be specified as "a successful frozen and banked umbilical stem cell transplantation in a boy ..." – corrected too.   

4) in figure 6 there is no correspondence between the percentages indicated in the previous figures (3 and 4). On the contrary, in figure 5, the percentages present in figures 3 and 4 are correctly reported. we ask to double check the correctness of figure 6 (genetic material 2017): I expect: "useful" NO 48.1% -52.8%, YES 27.5% -35.5%;
"useless" NO 1.8% -4.0%, YES 0.8% -1.7%, "no opinion" NO 29.0% -31.5%, YES 20.2% -20.5% ; "I haven't heard" NO 21.1% -11.7%, YES 51.4% -42.3%. Check whether the data relating to the "umbilical cord bank" are crossed with the "genetic material" data.

Answer: thank you very much for pointing this out. All data in figure 6 were wrong insterted. I corrected it and double-checked.

5) It is recommended to review the conclusions in the light of the re-evaluation of the data

Answer: Conclusions were reviewed. The percentages were at comparable level so the driven conclusions didn’t need re-writing.